# *DBP7* and *YRF1-6* Are Involved in Cell Sensitivity to LiCl by Regulating the Translation of *PGM2* mRNA

**DOI:** 10.3390/ijms24021785

**Published:** 2023-01-16

**Authors:** Sasi Kumar Jagadeesan, Mustafa Al-gafari, Jiashu Wang, Sarah Takallou, Danielle Allard, Maryam Hajikarimlou, Thomas David Daniel Kazmirchuk, Houman Moteshareie, Kamaledin B. Said, Reza Nokhbeh, Myron Smith, Bahram Samanfar, Ashkan Golshani

**Affiliations:** 1Ottawa Institute of Systems Biology, University of Ottawa, Ottawa, ON K1N 6N5, Canada; 2Department of Biology, Carleton University, Ottawa, ON K1S 5B6, Canada; 3Biotechnology Laboratory, Environmental Health Science and Research Bureau, Healthy Environments and Consumer Safety Branch, Health Canada, Ottawa, ON K1A 0K9, Canada; 4Department of Pathology and Microbiology, College of Medicine, University of Hail, Hail 55476, Saudi Arabia; 5Agriculture and Agri-Food Canada, Ottawa Research and Development Centre (ORDC), Ottawa, ON K1A 0C6, Canada

**Keywords:** molecular toxicity, lithium chloride, bipolar disorder, cell sensitivity, gene expression, translation, yeast

## Abstract

Lithium chloride (LiCl) has been widely researched and utilized as a therapeutic option for bipolar disorder (BD). Several pathways, including cell signaling and signal transduction pathways in mammalian cells, are shown to be regulated by LiCl. LiCl can negatively control the expression and activity of *PGM2*, a phosphoglucomutase that influences sugar metabolism in yeast. In the presence of galactose, when yeast cells are challenged by LiCl, the phosphoglucomutase activity of PGM2p is decreased, causing an increase in the concentration of toxic galactose metabolism intermediates that result in cell sensitivity. Here, we report that the null yeast mutant strains *DBP7*∆ and *YRF1-6*∆ exhibit increased LiCl sensitivity on galactose-containing media. Additionally, we demonstrate that *DBP7* and *YRF1-6* modulate the translational level of *PGM2* mRNA, and the observed alteration in translation seems to be associated with the 5′-untranslated region (UTR) of *PGM2* mRNA. Furthermore, we observe that *DBP7* and *YRF1-6* influence, to varying degrees, the translation of other mRNAs that carry different 5′-UTR secondary structures.

## 1. Introduction

Lithium is presently the first-line therapeutic choice for people suffering from bipolar illness. Bipolar disorder (BD), characterized by periodic bouts of severe depression and (hypo)mania interspersed with periods of remission, affects an estimated 1–5% of the adult population worldwide. Mood stabilizers, including anticonvulsants, atypical antipsychotics, and lithium salts, are often used to treat acute bouts of depression in the short and long term, as well as to prevent recurrences in the long run [1,2]. A lithium-based treatment regime for BD has been shown to be effective in avoiding mood relapses and lowering suicide probability in BD patients [3]. Approximately 30% of patients with BD are believed to be great responders to preventive medications, whereas 70% of patients exhibit varying degrees of response to Li [4]. LiCl is shown to alter the signaling pathways of protein kinase C and glycogen synthase kinase 3 and has immediate effects on neuroplasticity and behavior [5,6].

Previous work has revealed that when galactose is used as a sugar source, the budding yeast *Saccharomyces cerevisiae* is sensitive to LiCl exposure [7]. Alterations in *PGM2* expression and activity were shown to cause the observed LiCl sensitivity. *PGM2* is a phosphoglucomutase that aids galactose entry into the glycolysis process [7,8,9]. In the process of glycogenolysis, the conversion of glucose-1-phosphate to glucose-6-phosphate is facilitated by the *PGM2* enzyme. Impeding *PGM2* activity results in harmful intermediate metabolite aggregation and induces cell toxicity in yeast cells. As a result, yeast cell growth is substantially inhibited when grown on a galactose medium containing LiCl owing to metabolite buildup and glycolysis impairment [9]. LiCl has also been reported to disrupt the enzyme activity associated with ribosomal biogenesis, mRNA maturation in the cytoplasm, and rRNA processing, suggesting a potential translational inhibitory mechanism [10,11]. During the translation process, LiCl seems to alter the activity of certain eukaryotic translation initiation factors (eIFs), which are key elements in translation initiation regulation and function [8]. Translation initiation factor *eIF4A*, known as *TIF2* in yeast, is an archetypal DEAD-box RNA helicase which functions in tandem with *eIF4B*, *eIF4E, eIF4G,* and *eIF4H* and unwinds mRNA secondary structures at the 5’ untranslated region (UTR), in preparation for ribosomal binding. Ribosome scanning of the mRNAs is aided by *eIF4A* ATPase-dependent duplex-unwinding activity, and the latest findings suggest that this activity is dependent on both RNA secondary structures and sequence patterns [12]. *eIF4A* has been implicated in yeast’s response to LiCl stress [8]. Overexpression of *eIF4A* has been found to restore yeast cell sensitivity to LiCl in a galactose medium [13,14].

The mRNA 5′-UTR may include various regulatory elements, such as a translation initiation motif, upstream AUGs, upstream ORFs, 5′-cap structure, terminal oligo-pyrimidine strands, G-quadruplexes, internal ribosome entry sites, and secondary structures [15]. Among these, primarily the secondary structures at 5′-UTRs are thought to influence translation efficiency significantly [16]. The presence of a strong secondary structure in the 5′-UTR of an mRNA can substantially reduce translation efficiency by extending the “dwell time” of preinitiation complex formation during translation initiation [17]. Recent investigations have established a connection between LiCl cell sensitivity and the translation of structured mRNAs in yeast [13,14,18].

Here, we observe that the yeast mutant strains *dbp7*∆ and *yrf1-6*∆ cultured on a galactose medium had enhanced cell sensitivity to LiCl. *DBP7* exhibits helicase activity and is reported to be actively involved in ribosomal biogenesis [19]. *YRF1-6* encodes a DNA helicase called Y-Helicase protein 1 and shares homology with *eif4A* [20]. Our findings suggest that *DBP7* and *YRF1-6* influence *PGM2* translation. This observed activity appears to impact the structured 5’-UTR of *PGM2* mRNA in addition to many additional structured 5’-UTRs.

## 2. Results

### 2.1. DBP7 and YRF1-6 Gene Deletions Diminish Yeast Tolerance to Lithium Chloride

The functional characteristics of various chemicals and biologically active compounds can be studied using chemical genetic methods [21,22]. These approaches provide imperative information on the fundamental mechanism of action of a compound as well as its secondary mode of action inside a cell. In this context, the sensitivity of gene mutant strains to a targeted drug serves as a powerful tool for identifying cellular target pathways for that drug. In this research investigation, we discovered that two gene deletion mutants, *DBP7* and *YRF1-6,* were more sensitive to LiCl (Figure 1B,C) than a WT control strain. Deletion of *DBP7* and *YRF1-6* significantly reduced cell growth in galactose-containing media supplemented with 10 mM LiCl. This suggests a functional relationship between these genes and the LiCl mechanism of action on yeast cells. It remains possible that the observed sensitivity for the gene deletion mutants might be due to an unintended secondary mutation within these strains. To investigate that possibility, we further established a clear connection between the observed impairments from LiCl sensitivity and our potential gene targets by showing that the re-introduction of the deleted genes back into the deletion strains repaired the previously impaired growth. (Figure 1B). Our quantitative study validates these results by directly comparing the colony-forming units in a medium containing 10mM LiCl (Figure 1C). In comparison to the control strain, gene deletion mutants formed fewer colonies, indicating an increased sensitivity to LiCl. Deletion of *TIF2, DBP7*, and *YRF1-6* resulted in a decrease in the formation of colonies, as seen in Figure 1C. As shown before, restoration of the deleted genes back into the target gene mutant strains elevated colony formation to the control levels. The cells exhibited no discernible sensitivity when we evaluated our target strains on the YPgal medium without LiCl (Figure 1A). We also investigated how yeast strains responded to LiCl, with glucose being the carbon source. When glucose was used, there was no change in sensitivity in yeast mutant strains and WT, as expected (Appendix A). 

It is thought that LiCl inhibits *PGM2* expression in the presence of galactose, causing yeast cells to accumulate toxic intermediate metabolites during galactose metabolism. In the initial stages of galactose metabolism, *GAL1* encodes galactokinase, which is involved in the phosphorylation of α-D-galactose to α-D-galactose-1-phosphate in yeast. To examine the involvement of *DBP7* and *YRF1-6* in LiCl sensitivity when galactose metabolism is disrupted, we created *DBP7* and *YRF1-6* double gene deletions in conjunction with the *GAL1* gene. Interestingly, *GAL1* double mutant cells with our candidate genes were no longer sensitive to LiCl, suggesting that the observed sensitivity for *DBP7* and *YRF1-6* deletion mutants is connected to galactose metabolism (Figure 1B,C). Notably, *DBP7* and *YRF1-6* have no reported connection to LiCl sensitivity or the biochemical pathways that might support these observations, making them intriguing gene candidates for further research.

### 2.2. DBP7 and YRF1-6 Regulate PGM2 Expression at the Translational Level

*PGM2* is a key target for LiCl sensitivity. We investigated the impact of *DBP7* and *YRF1-6* on the expression of *PGM2*. For protein content analysis, anti-GFP antibodies were used in western blot analysis to measure GFP-tagged PGM2p (Figure 2A). Under control conditions (no exposure to LiCl), gene deletions for *DBP7* and *YRF1-6* showed similar protein levels for PGM2p to that of the control strain. Intriguingly, after exposure to LiCl, the deletion of our target genes significantly reduced PGM2p protein levels in comparison to the WT. In addition, qRT-PCR was performed to investigate the impact of *DBP7* and *YRF1-6* gene deletions on the *PGM2* mRNA levels. As indicated in Figure 2B, there were no statistically notable variations in *PGM2* mRNA content between the mutant strains and the WT in the treatment and control groups. Therefore, it seems that the deletion of *DBP7* and *YRF1-6* has little impact on the levels of *PGM2* mRNA. As a result, *DBP7* and *YRF1-6* seem to control PGM2p protein expression. These observations support our previous research findings, indicating that the deletion of specific genes that conferred LiCl sensitivity caused a reduction in *PGM2* translation in the presence of LiCl [13,18].

### 2.3. DBP7 and YRF1-6 Deletion Directly Impact β-Galactosidase Reporter mRNAs with Secondary Structures at 5′-UTR

Previously, it was shown that *PGM2* mRNA at the 5’-UTR was utilized by multiple genes to regulate the expression of *PGM2* at the translational level [10,13]. The *PGM2* mRNA at the 5’-UTR is predicted to have a stem-loop region (Appendix A). Expression of *PGM2* was significantly diminished with the removal of *TIF2*, a helicase protein that unwinds mRNA strands during the initiation of translation [10,14]. Using the secondary structure of *PGM2* mRNA at the 5’-UTR, we next asked whether *DBP7* and *YRF1-6* had an influence on *PGM2* translation through this specific sequence. For this purpose, a *lacz* expression cassette in the p416 expression plasmid with *PGM2* 5’-UTR incorporated in front of the *Lacz* gene was used. The parental p416 plasmid that lacked a secondary structure in front of the *lacz* gene (control plasmid), and the pPGM2 plasmid that carried the *PGM2* hairpin structure at its 5′-UTR, were transformed into our target yeast mutant strains, as well as the WT strain. *Lacz* gene expression was measured using *β-galactosidase* activity (Figure 3A). As anticipated, we found no apparent change in *β-galactosidase* activity among our target strains harboring the parental plasmid (p416), in which the mRNAs lacked a structure at their 5’-UTR. However, *β-galactosidase* activity was considerably reduced in *dbp7∆*, and *yrf1-6∆,* similar to the *tif2∆* control, carrying pPGM2 expression plasmid that contains *PGM2* mRNA secondary structure at the 5’-UTR, suggesting a relationship between *DBP7* and *YRF1-6* activity and structured *PGM2* mRNA translation.

The impact of *DBP7* and *YRF1-6* on other structured mRNAs was then investigated. A total of four additional constructs were used in this experiment, each of which carried a unique structure at their 5’-UTR in front of a *lacz* reporter gene. pBcell has a structure derived from *BCL2* mRNA with a ∆G value of −20 kcal/mol. pRTN contains a structured 5’-UTR from *RTN4IP1* mRNA with a ∆G value of −29.8 kcal/mol, and the pTAR structure is obtained using *HIV1* mRNA with a ∆G value of −57.9 kcal/mol. The final construct, p2hair, has a synthetic structure with a high degree of complexity, ∆G = −33 kcal/mol. Using these constructs, we observed that deleting *DBP7* and *YRF1-6* considerably lowered the *lacz* expression carrying all four highly structured mRNAs at their 5′-UTR compared to the WT (Figure 4). This indicates that our target genes *DBP7* and *YRF1-6* may play a broader role in the translation, specifically in the translation of structured mRNAs. 

### 2.4. Deletion Mutants for DBP7 and YRF1-6 Show Increased Sensitivity to Phenethyl Isothiocyanate

It has been observed that phenethyl isothiocyanate (PEITC), a naturally occurring component of cruciferous vegetables, inhibits cap-dependent translation, specifically the translation of mRNAs harboring secondary structures at 5’UTR [23,24]. If *DBP7* and *YRF1-6* are affecting the translation of structured mRNAs, it might be expected that their gene deletion mutants may also possess increased sensitivity to PEITC. To study this, we subjected gene deletion mutants for *DBP7* and *YRF1-6* to sensitivity analysis using PEITC. It was observed that the deletion mutants of *DBP7* and *YRF1-6* had increased sensitivity to 15 μM PEITC (Figure 5). As expected, the re-introduction of the deleted genes into their respective target mutant strains reverted the observed sensitivity induced by PEITC treatment, indicating that the exhibited phenotypes are a direct consequence of the deletion of the target genes.

### 2.5. Genetic Interaction Analysis Connects the Activity of DBP7 and YRF1-6 to Protein Biosynthesis

The concept underlying genetic interaction (GI) analysis is that parallel pathways allow flexibility and tolerance to random damaging mutations, hence sustaining cell survival and homeostasis. In certain instances, a gene from one pathway may complement a gene from another. Therefore, when two genes in parallel pathways are eliminated, the fitness of the cell may drastically decline (cell sickness), or in severe cases, the cell may die (cell lethality). Because of the decreased cell fitness observed in the double mutants, this type of GI is referred to as a negative genetic interaction (nGI). nGIs have been commonly used in different studies to examine various functions for genes, identify complex biological networks and understand various molecular pathways [19,20,21].

To study GIs in yeast, two mating types: α-mating type (Mat α) and a-mating type (Mat a), are commonly utilized. Mat “α” contains the target gene deletion and is crossed with an array of Mat “a” single gene deletion to create double gene deletions [25]. The phenotypic fitness of the strains is measured using colony size. We used this approach to look for genetic links between our target genes, *DBP7* and *YRF1-6*, and nearly 1000 other genes, including ~700 linked to gene expression pathways and a random sample group of ~300 genes used as control (See Appendix A). We uncovered a series of intriguing nGIs as well as multiple shared gene hits between *DBP7* and *YRF1-6* (See Appendix A). A substantial proportion of these hits are linked with translation and translation initiation, as determined using the functional enrichment analysis (Figure 6).

For *DBP7*, we identified nGIs with genes including *ANB1*, *EBS1*, and *GCN3,* among others (Figure 6). *ANB1* is a ribosome-binding protein that encodes the eukaryotic translation initiation factor *eIF5A* [26]. It actively catalyzes the formation of peptide bonds and facilitates translation by resolving ribosomal stalls during protein biosynthesis [27]. *GCN3*, the alpha subunit of the translation initiation factor *eIF2B*, is primarily engaged in translation initiation, mediating the exchange of guanine nucleotides associated with GTPases, thereby enabling the formation of preinitiation complex in translation initiation [28,29]. Another intriguing interactor, *EBS1,* is engaged largely in translation inhibition and nonsense-mediated decay [30]. It physically interacts with the cap-binding proteins cdc33p and Nam7p, which contribute to translation initiation and control [30,31].

Similarly, *YRF1-6* interacted with *TMA64*, *BUD27*, and *eIF2A*, among others, that are associated with translation (Figure 6). *TMA64* is linked to translation initiation, encodes an RNA binding domain, and interacts with small ribosomal subunits [32]. It has been demonstrated that *TMA64* influences protein synthesis, and its homologous protein in mammals, *eIF2D*, facilitates the attachment of the 43S initiation complex to mRNA, forming a 48S initiation complex during translation initiation [33]. *BUD27* is a cytoplasmic protein that aids in the development of the preinitiation translation complex [34]. *eIF2A* encodes the translation initiation factor *eIF2a*, which is associated with both 40S and 80S ribosomal subunits [34,35]. It is noteworthy that *eIF2a* has previously been reported to influence mRNA secondary structures at the 5′-UTR [36].

Numerous common interactors between *DBP7* and *YRF1-6* were also discovered, including *SLH1*, *SHE3*, and *PUF6*. *SLH1* codes for a putative RNA helicase that has been linked to the inhibition of translation of non-poly(A) mRNAs [37,38]. *SHE3* codes for an adapter protein involved in mRNA localization and protein accumulation and promotes general translation [39,40]. The protein encoded by *PUF6* binds to the 3’UTR of *ASH1* mRNA and plays a critical role in the synthesis of the 60S ribosomal subunit [41,42].

Conditional nGI occurs when a specified condition is met, enabling an interaction. Such conditions may include nutrient deprivation/starvation, exposure to cold or heat shock, or the occurrence of a bioactive chemical at a sub-inhibitory concentration. They indicate functional connections between genes that develop in response to specific environments [43,44]. For instance, under specific conditions such as DNA damage, certain gene function(s) may be modified, and these varied activities are functionally connected to the activity of an interacting partner. Such case-dependent relationships constitute the foundation of conditional nGIs [45]. In our analysis, with a sub-inhibitory drug concentration of LiCl (3 mM), we evaluated nGIs for *DBP7* and *YRF1-6* (See Appendix A). The nGIs observed in this circumstance are formed in response to exposure to LiCl. As illustrated in Figure 7, we found new nGIs for our candidate genes *DBP7* and *YRF1-6*.

Under LiCl treatment, *DBP7* is associated with numerous translation control genes, including *DTD1*, *CTK1*, and *EAP1,* among others. These interactions were not observed in the absence of LiCl. *DTD1* is involved in the regulation of protein synthesis machinery under nutrient deprivation or stress, and it affects nonsense suppression via alteration of the protein translation machinery [46]. *CTK1 regulates* various translation processes, such as mRNA processing, ribosomal binding, and initiation complex formation, as well as functioning as a vital factor in enhancing translation fidelity. It also stimulates the formation of the 80S initiation complex [47,48,49]. Another intriguing negative interactor is *EAP1*, which encodes an *eIF4E*-associated binding protein and controls the global translation rate by inhibiting eIF4G-eIF4E binding [50,51]. It stimulates decapping and accelerates mRNA degradation by promoting association with eIF4E [52]. Of interest, eIF4E is a component of the eIF4F complex, which also includes eIF4A, eIF4G, and eIF4E.

In addition, *YRF1-6* interacted conditionally with several translation control genes, including *EDC1* and *PSK1*. *EDC1* encodes for an RNA binding protein that controls mRNA decapping and plays an active role in translation under stress conditions [53,54]. *PSK1* encodes for a kinase with a PAS domain that controls protein synthesis during glycogen formation and is primarily involved in the process of protein phosphorylation [55]

Furthermore, we identified several common conditional interactors between *DBP7* and *YRF1-6* that are involved in translation regulation, including *SRO9, GIS2,* and *DBP1,* among others. *SRO9* encodes a cytoplasmic RNA-binding protein that controls the global translation rate and is involved in mRNA binding and translation regulation. It is found in polysomes and cytoplasmic stress granules [56,57]. *DBP1* encodes for a DEAD box protein that interacts with the 48S preinitiation complex during translation initiation and recruits mRNAs with structured 5′-UTRs [58,59]. *GIS2* is involved in translation control by regulating mRNA binding and localization, and it also influences mRNA degradation [60].

Phenotypic suppression array (PSA) analysis explores a different type of interaction where the overexpression of a target gene compensates for the absence of another, under a specific condition [22,61,62,63]. It is an informative type of interaction because it may reveal functional associations between two genes. In this analysis, the gene mutant array was subjected to 10 mM LiCl. Several tested mutant strains demonstrated greater cell sensitivity. We then sought to compensate for the reported sensitive phenotypes using the introduction of *DBP7* and *YRF1-6* overexpression plasmids. As mentioned above, colony size was used to measure phenotypic fitness. The incorporation of *DBP7* or *YRF1-6* overexpression plasmid recovered the cell sensitivity caused by LiCl treatment in four gene deletion mutant strains *MRN1, GCN2, BCK1,* and *DHH1* (Figure 8). *MRN1* encodes an RNA-binding protein that regulates translation by binding to specific categories of mRNA that include uORFs and IRES motifs [64,65]. *GCN2* encodes for a protein kinase found in cytosolic ribosomes that controls translation initiation by phosphorylating the translation initiation factor eIF2 and influences the overall translation rate [66]. *BCK1* encodes for a kinase which influences several cellular processes, including translation [67]. *DHH1* is a cytoplasmic DEAD-box helicase that facilitates mRNA stability, mRNA degradation, and polyadenylation at the 5′-UTR of mRNAs [68].

## 3. Discussion

It is well-documented that mRNA structures at the 5’-UTR can affect translation initiation activity [15]. The findings in this study demonstrate that *DBP7* and *YRF1-6* regulate *PGM2* mRNA translation at its 5′-UTR region. Similar findings were found for additional mRNAs that have distinct secondary structures at their 5′-UTRs. Several hypotheses may be suggested to explain the activity of *DBP7* and *YRF1-6* on structured mRNA translation. The simplest interpretation is that one or both proteins may contain helicase function, implicated in the 5′-UTR unwinding of structured mRNAs. This is a likely explanation as these genes are already thought to have helicase functions [69,70]. Another possibility is that these factors may also influence the activity of other helicases and by doing so, they alter structured mRNA translation. In addition, it is possible that *DBP7* and *YRF1-6* could modify ribosome function, which may have an effect on the translation of structured mRNAs. Consequently, it is possible that in the absence of *DBP7,* ribosomes might have a slight adjustment making them less amenable to translating structured mRNAs. In agreement with this possibility, we also observed several interactions for *DBP7* and *YRF1-6* with various ribosomal proteins in the current study, indicating a possible relationship between our candidate genes and translation machinery. An alternate argument is that these genes may influence the biology of mRNAs. *DBP7* and *YRF1-6* may thereby influence mRNA biosynthesis, which may affect the translatability of structured RNAs.

Interestingly deletion of the target genes did not show similar patterns in the levels of translation reduction for the different constructs, pBCell, pRTN, pTAR, and p2hair (Figure 4). For example, the deletion of *DBP7* reduced the translation of pBCell and p2hair by approximately 90%. However, its lowering effect on the translation of pRTN was approximately 40%. Similarly, deletion of the positive control, *TIF2*, appeared to affect the translation of pBCell more than the other constructs. Each of the constructs carry distinct secondary structures at the 5′-UTR that may explain these differences (Figure 4). These structures vary in configuration, free energy value, loop sequence, GC content, length, and their distance from the 5’ cap, to name a few. Such differences are thought to be important in restricting translation activity [71]. For subsequent research, it would be interesting to study the causes of the observed differences in the translation of different constructs using particular gene deletion mutants.

The discovery of new gene functions associated with the regulation of structured mRNA translation reveals that structured mRNA translation is more mosaic and complicated than previously assumed. Moreover, it demonstrates that mRNAs with complex structures may have increased activity in dynamic regulatory networks and biological pathways during translation. Depending on the structure and content of these sequences, different host mutations may result in different mRNA translation levels for different hosts. In addition, the present study provides an additional link to LiCl’s influence on biological function and regulation of structured mRNA translation in yeast. Translational or other pre-clinical research are warranted to discover how these mechanistic insights could improve the responders and non-responders of bipolar disease patients to LiCl therapy. Investigating the 5′-UTR of different human genes may uncover new mechanisms for LiCl activity. Due to the therapeutic use of LiCl, subsequent investigations on the impact of other compounds affecting the translation of complex mRNA secondary structures at the 5′-UTR would be of interest.

## 4. Materials and Methods

### 4.1. Strains and Plasmids

Deletion strains in BY4741 background (MATa orfΔ: kanMX4 his3Δ1 leu2Δ0 met15Δ0 ura3Δ0) were obtained from the yeast gene deletion collection [72] and verified using PCR analysis. Deletion strains in BY4742 background (MATα can1Δ:STE2pr-HIS3 lyp11Δ ura31Δ leu21Δ his31Δ met151Δ) were generated using PCR-mediated gene disruption as previously described [20,73]. Overexpression plasmids were acquired from the yeast overexpression plasmid library [20]. PGM2p-GFP fusion strain was obtained from the Yeast gene-GFP fusion library [21] and verified using PCR analysis. The control plasmid, p416, carried a *lacz* expression cassette with a gal promoter [74]. The HIV1 (*TAR1*), RTN (*FOAP-11*), Bcell (*BCL-2*), 2-hair, and *PGM2* hairpin constructs were cloned by utilizing the XbaI restriction site, located upstream of *lacz* ORF and in between the gal promoter and the *β-galactosidase* reporter gene in the p416 expression vector (*lacz* expression cassette).

pTAR construct contains 5′-UTR of HIV1-*TAR* gene (5’GGTTCTCTGGTTAGCCAGATCTGAGCCCGGGAGCTCTCTGGCTAGCTAGGGAACCCATGCTTAAGCCTCAATAAAGCTTGCCTTGAGTGCTTCAAGTAGTGTGTGCC 3’), pRTN has 5′-UTR of *FOAP-11* gene(5’GGGATTTTTACATCGTCTTGGTAAAGGCGTGTGACCCATAGGTTTTTTAGATCAAACACGTCTTTACAAAGGTGATCTAAGTATCTC3’), pBCell carries 5′-UTR of *BCL-2* gene (5′ GGGGGCCGUGGGGUGGGAGCUGGGGGGGCCGUGGGGU GGGAGCUGGG 3′), 2-hair contains 5′-UTR of the construct (5’ CTTGGTAAAGGGGGUGGTCTGAGCCCGGGAGCTCTCTGCTGCTTAAGCCTCGGATTTT 3’), and pPGM2 construct has 5′-UTR of *PGM2* gene (5′TAATAAGAAAAAGATCAC CAATCTTTCTCAGTAAAAAAAGAACAAAAGTTAACATAACAT 3′). Furthermore, all plasmids carried an ampicillin-resistant gene for selection during plasmid transformations in the DH5α strain of *Escherichia coli* (*E. coli*). *URA3* gene was used for selecting transformed yeast. Plasmid extraction of *E. coli* was conducted with the GeneJET plasmid miniprep kit (Thermofisher^®^, Mississauga, ON, Canada), and yeast plasmid extraction was performed using the yeast plasmid miniprep kit (Omega Biotek, Norcross, GA, USA). Yeast colonies were grown in a YP medium containing 1% yeast extract and 2% peptone. In yeast media, the carbon source was 2% galactose or 2% glucose. For solid media, 2% agar was utilized. To prepare complete synthetic media with selective amino acids, a 0.67% yeast nitrogen base without amino acids and a 0.2% amino acid dropout mix were used. *E. coli* was cultured in an LB medium (Lysogeny Broth).

### 4.2. Drug Sensitivity Analysis

Specific yeast colonies were cultured for 48 h at 30 °C in liquid YPgal (YP + 2% galactose) media. Spot test analysis was performed by spotting serial dilutions of cell suspensions onto solid media with or without LiCl. For drug sensitivity analysis to LiCl, galactose media or glucose media containing 10 mM LiCl concentration was used, as discussed previously [13,18]. The ssensitivity of our target strains to LiCl was determined by examining the growth of gene deletion mutants compared to those of the wild type (WT) strain. To validate that the reported sensitivities correspond to the deletion of our candidate genes, overexpression constructs pDBP7 and pYRF1-6 were transformed into the target gene deletion strains. For colony count analysis, 100 μL of each strain at 10^−4^ cell culture concentration was plated onto YPgal petri plates with or without LiCl. The plates were evaluated based on colony formation after 48 h. All the experiments were carried out in triplicates. To evaluate statistical significance, one-way ANOVA analysis (*p*-value ≤ 0.05) was performed. For Phenethyl Isothiocyanate (PEITC) sensitivity analysis, liquid cultures were normalized to OD 0.1 and plated onto a YPD medium with 15 µM PEITC.

### 4.3. mRNA Quantification Analysis (qRT-PCR)

The *PGM2* mRNA levels were determined using a PGM2p-GFP yeast strain cultured in liquid YPgal medium overnight with or without 10 mM LiCl, as previously described [14,18]. The Qiagen^®^ RNeasy Mini Pack (Qiagen, Toronto, ON, Canada) was used to harvest total RNA. The complementary DNA (cDNA) was synthesized with the help of a Bio-Rad^®^ cDNA Synthesis Kit, (Bio-Rad^®^, Mississauga, ON, Canada) and the quantitative real-time PCR was carried out with Bio-Rad^®^ IQ SYBR Green Supermix (Bio-Rad^®^, Mississauga, ON, Canada). As an internal control, the housekeeping gene *PGK1* was employed. Methodology and data analysis were conducted per MIQE principles [75]. At least three technical and biological replicates were used in each experiment. qPCR primers used in this study are as follows:

*PGK1* Forward: ATGTCTTTATCTTCAAAGTT; Reverse: TTATTTCTTTTCGGATAAGA; *PGM2* Forward: GGTGACT CCGTCGCAATTAT; Reverse: CGTCGAACAAAGCACAGAAA.

### 4.4. Western Blot Analysis

PGM2p-GFP fusion protein content was analyzed using quantitative western blotting, as indicated previously [13,14]. Gene deletion mutant strains in the PGM2p-GFP background were cultured in media either containing or lacking LiCl to evaluate the protein levels of *PGM2.* Protein concentration was determined using the Bradford Protein Assay (BSA). Using Mini-PROTEAN Tetra cell electrophoresis equipment (Bio-Rad^®^, Mississauga, ON, Canada), 50 µg of total extracted protein was run on a 10% SDS-PAGE gel. Trans-Blot Semi-Dry Transfer (Bio-Rad^®^, Mississauga, ON, Canada) was used to transfer protein bands onto a 0.45 m nitrocellulose membrane. A mouse monoclonal anti-GFP antibody (Santa Cruz^®^) was used to detect PGM2p-GFP protein levels, and a mouse monoclonal anti-PGK1 antibody was used to assess PGK1 protein levels (internal control). At least three technical and biological replicates were used in each experiment.

### 4.5. Quantitative β-Galactosidase Assay

To assess the activity of *lacZ* expression cassettes, a quantitative *β-galactosidase* assay experiment was conducted using ONPG (O-nitrophenyl—D-galactopyranoside), as reported [14,18]. All experiments were conducted in triplicates, and the one-way ANOVA method was used to assess statistically significant changes.

### 4.6. Genetic Interaction Analysis

Synthetic Genetic Array (SGA) analysis was conducted by mating two strains, MATa mating type and MATα mating type, to produce progenies carrying both gene deletions. MATa strains were obtained from the yeast knockout library [72]. MATα mating strains that carried the target gene knockout were generated using homologous recombination, as previously reported [20,72]. PCR analysis was used to confirm a successful gene knockout. We investigated genetic interactions between our target genes (GI) using a 384-formatted SGA, as described before [25]. Conditional SGA was carried out by introducing the double mutants to a low sub-inhibitory concentration of LiCl (3 mM) [61]. Phenotypic Suppression Array (PSA) analysis was performed by mating MATa single deletion array with a MATα yeast strain containing the target overexpression plasmid [22,61]. Compensation for LiCl sensitivity of the single deletion mutants with the overexpression analysis was used to establish a putative functional relationship between the deleted and the overexpressed gene as before two genes [19,33].

Cell fitness was determined using colony size measurements [28,34]. SGA Software was used to determine the colony size and similarity of colonies [76]. A Cell Fitness reduction of 30% or more was considered. Each experiment was conducted three times, and those findings that were consistent in at least two repeats were considered. Using Gene ontology enrichment methods, observed gene hits were grouped according to their biological and molecular functions using Genemania. http://genemania.org (Accessed on 20 December 2021).

## Figures and Tables

**Figure 1 ijms-24-01785-f001:**
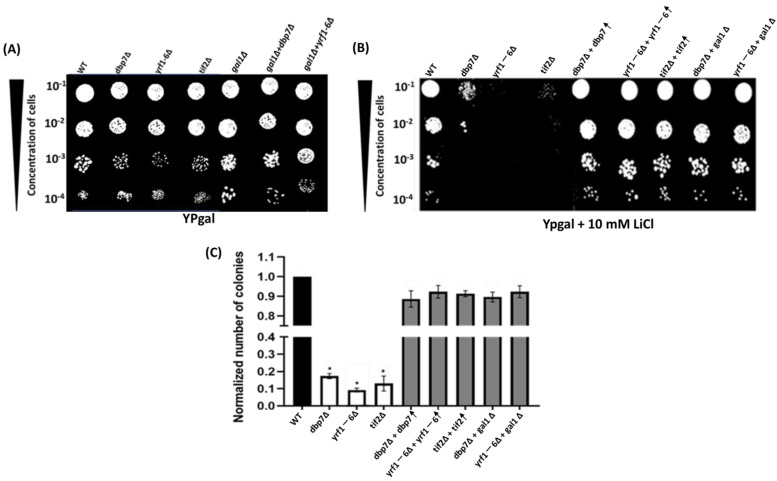
Spot test and colony count analysis demonstrated that *dbp7*∆ and *yrf1-6*∆ exhibit increased LiCl sensitivity when compared to the WT. In (**A**,**B**), yeast cells were serially diluted (10^−1^ to 10^−4^) and spotted onto Ypgal media with or without 10mM LiCl. *Dbp7*∆ and *Yrf1-6*∆ exhibit less growth when exposed to LiCl. In gene deletion mutants, restoration of the target genes resulted in a recovery of LiCl sensitivity. Similarly, *GAL1* deletion reversed LiCl sensitivity in target gene deletion mutants. WT and *tif2*∆ are used as controls. *(***C**) When treated with 10mM LiCl, mutant strains *dbp7*∆ and *yrf1-6*∆ formed considerably fewer colonies than the WT in the colony count analysis. The bars reflect mean values (*n* ≥ 3), and the error bars represent standard deviation and the ‘*’ indicates statistically notable outcomes compared to the WT. Each experiment was conducted in triplicates, with similar outcomes.

**Figure 2 ijms-24-01785-f002:**
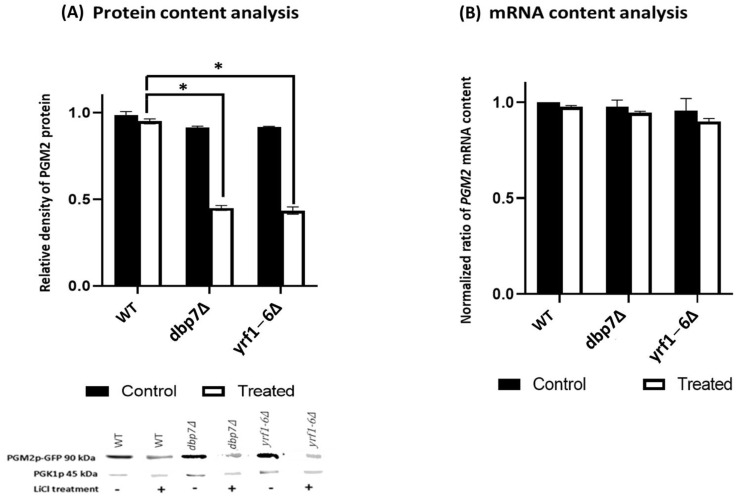
PGM2p-GFP protein and *PGM2* mRNA content were examined using western blot and qRT-PCR with and without LiCl treatment (**A**) In the deletion strains DBP7 and YRF1-6, the amount of PGM2p-GFP protein is reduced by LiCl treatment. As an internal control, the housekeeping protein PGK1p was employed, and the data was standardized based on it. (**B**) There was no statistically significant variation in mRNA content across the yeast variants studied. Individual Ct values in this experiment were between 20 and 22.5. *PGK1* mRNA was used as an internal control. All experiments were carried out in triplicates. The bars reflect the mean values (*n* ≥ 3), the error bars represent the standard deviation, and the ‘*’ indicates statistically notable outcomes compared to the WT.

**Figure 3 ijms-24-01785-f003:**
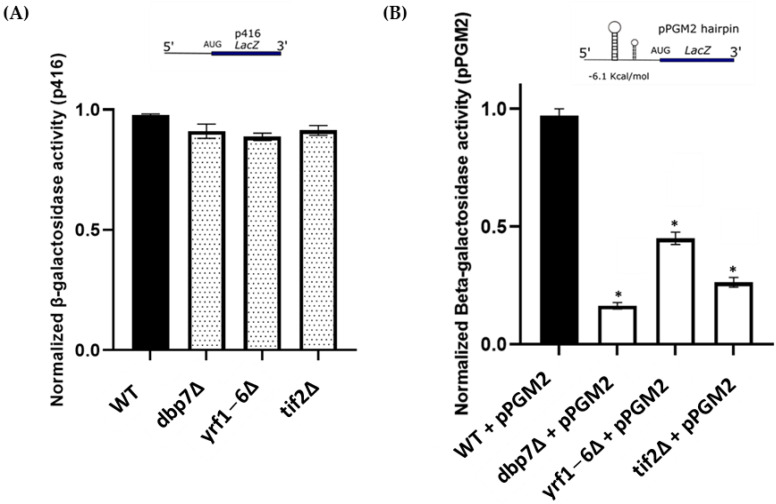
*β-galactosidase* analysis of our candidate gene deletions carrying the construct p416, with and without *PGM2* hairpin. (**A**) Yeast strains carrying intact p416 construct lacking a secondary structure upstream of the *lacz* reporter mRNA. *β-galactosidase* activity observed for the yeast mutant strains and WT had no significant difference. (**B**) Yeast strains carrying a modified p416 construct with *PGM2* mRNA 5′-UTR structure upstream of the *lacz* reporter mRNA. *Β*-*galactosidase* activity was reduced in the *tif2*∆, *dbp7*∆, and *yrf1-6*∆ strains compared to the WT. Bars represent mean values (*n* ≥ 3), and ‘*’ represents statistically significant results compared to the WT. The insets illustrate schematic reporter mRNA structures.

**Figure 4 ijms-24-01785-f004:**
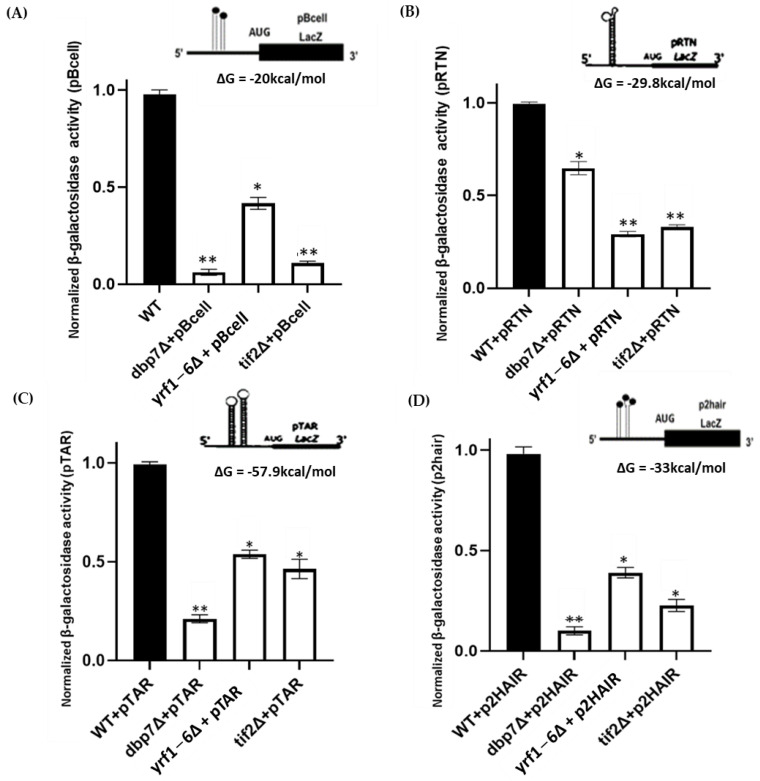
Analysis of *β-galactosidase* activity in yeast strains. (**A**) Activities of *β-galactosidase* mRNAs with a strong hairpin construct, pBcell carrying the 5’-UTR of the *BCL-2* gene located upstream of a *lacz* reporter gene, were dramatically reduced in yeast mutant strains relative to the WT. The pRTN (**B**) and pTAR (**C**) constructs, respectively, contain the structures from the 5’-UTR of the *FOAP-11* and HIV *TAR-1* genes in front of the *β-galactosidase* reporter mRNA and have considerably decreased values in strains *dbp7*∆, *yrf1-6*∆, and *tif2*∆ compared to the WT. The construct p2hair (**D**) carries two strong synthetic hairpin structures in front of the *β-galactosidase* mRNA. Similarly, the *β-galactosidase* activity of p2hair was reduced in mutant strains *dbp7*∆, *yrf1-6*∆, and *tif2*∆ compared to the WT. The values were normalized to the WT. All experiments were carried out in triplicates (*n* ≥ 3), and the error bars reflect the standard deviation. ‘*’ and ‘**’ reflect statistically significant results compared to WT, with *p*-values ≤ 0.05 or ≤ 0.005, respectively. The insets illustrate schematic reporter mRNA structures.

**Figure 5 ijms-24-01785-f005:**
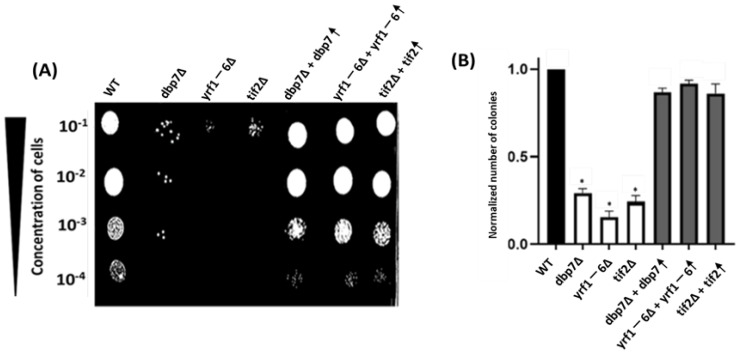
Spot test and colony count analysis demonstrated that *dbp7*∆ and *yrf1-6*∆ exhibit increased PEITC sensitivity. (**A**) Yeast cells are serially diluted (10^−1^ to 10^−4^) and spotted onto YPD media with 15 μM PEITC. *tif2*∆, *dbp7*∆, and *yrf1-6*∆ demonstrated enhanced sensitivity to 15 uM PEITC drug treatment relative to the WT, which is reversed by reintroducing corresponding overexpression plasmids. All experiments were conducted in triplicates with similar outcomes. (**B**) When treated with PEITC, the mutant strains *tif2*∆, *dbp7*∆, and *yrf1-6*∆ formed considerably fewer colonies than the WT. Yeast strains bearing the relevant overexpression plasmids had PEITC sensitivities that resembled that of the WT. The bars reflect mean values (*n* ≥ 3), and the error bars represent standard deviation; One-way ANOVA with Post Hoc Tukey Test, *p*-value ≤0.05, was used to establish statistically significant observations indicated by ‘*’.

**Figure 6 ijms-24-01785-f006:**
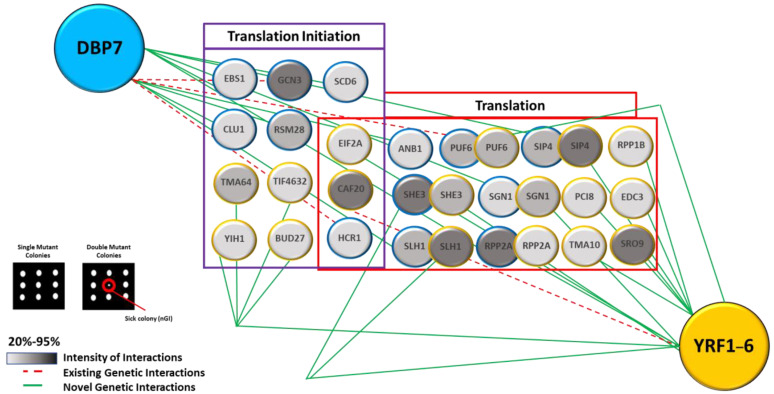
Negative genetic interactions (nGIs) for *DBP7* and *YRF1-6*. A group of interactors belongs to the category of translation for *DBP7* (*p* ≤ 3.4 × 10^−6^) and *YRF1-6* (*p* ≤ 5.8 × 10^−9^). Several interactors also fall into the category of translation initiation for *DBP7* (*p* ≤ 2 × 10^−8^) and *YRF1-6* (*p* ≤ 1.9 × 10^−6^). Mutual hits shared by two target genes include *SHE3*, *PUF6*, *SLH1*, *SGN1*, *SIP4*, and *RPP2A*. Genes are represented as circles (nodes), while nGIs are represented by lines/dotted lines (edges). A representative nGI is indicated in the inset by the red circle. The strength of interactions is illustrated by gradients of color, with darker shades indicating lower fitness. GIs that were previously documented in the literature are referred to as existing GIs.

**Figure 7 ijms-24-01785-f007:**
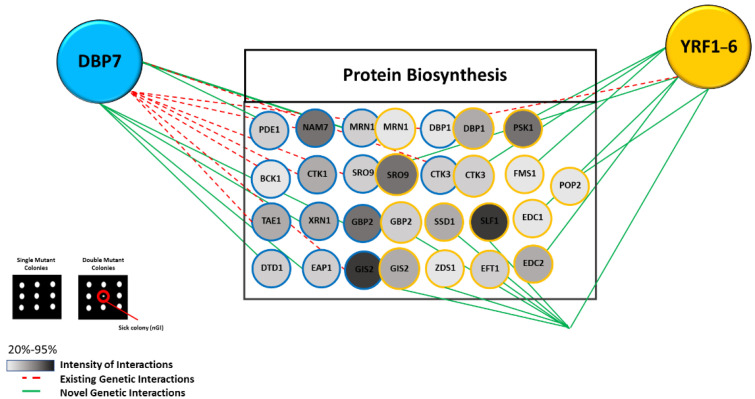
Conditional nGIs for *DBP7* and *YRF1-6* under a subtle sub-inhibitory concentration of LiCl (3 mM). A collection of genetic interactors implicated in the control of translation for both *DBP7* (*p* ≤ 8.1 × 10^−6^) and *YRF1-6* (*p* ≤ 6.2 × 10^−6^) is observed. *DBP7* and *YRF1-6* share common interactors *SRO9*, *GIS2*, *DBP1*, *CTK3*, *GBP2*, and *MRN1*. Genes are represented as circles (nodes), while nGIs are represented by lines/dotted lines (edges). A representative nGI is indicated in the inset by the red circle. The strength of interactions is illustrated by gradients of colour, with darker shades indicating lower fitness. GIs that were previously documented in the literature are referred to as existing GIs.

**Figure 8 ijms-24-01785-f008:**
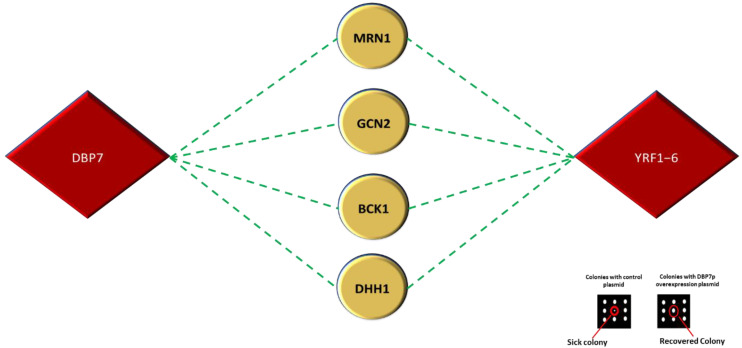
Overexpression of *DBP7* and *YRF1-6* restores the cell sensitivity caused by 10 mM LiCl treatment in yeast mutant strains *mrn1*∆, *gcn2*∆, *bck1*∆, *and dhh1*∆ to 10 mM LiCl. *MRN1, GCN2, BCK1,* and *DHH1* are involved in the regulation of translation. A representative interaction is indicated in the inset by the red circle; it illustrates the phenotype recovered using the introduction of *DBP7* overexpression plasmid in media supplemented with 10 mM LiCl.

## Data Availability

All data generated and/or analyzed during this study are included in this research article and/or its Appendix A.

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
