# Peer review of "DBP7 and YRF1-6 Are Involved in Cell Sensitivity to LiCl by Regulating the Translation of PGM2 mRNA"

_ijms, 2023, doi:10.3390/ijms24021785_

Round 1
Reviewer 1 Report
Please see the attached file.

Reviewer 2 Report
Review comments
In this study, the anthors reported that the deletion of DBP7 and YRFI-6 increases cell sensitivity to LiCl when the mutant strains were grown in a galactose medium, suggesting that DBP7 and YRFI-6 were involved in the regulation of PGM2 mRNA translation.
The research idea is relatively innovative, the research design is scientific and reasonable, the experimental data is authentic, the data statistical analysis method is reliable, and the language expression is smooth and easy to read. The research results have certain scientific and practical significance. The main problems are as follows.
Q1: LiCl sensitivity analysis for target strains in YPD media. The authors chose two doses of 10mM and 100mM. My concern is whether the researchers have used lower doses? Even in the pre experiment.
Q2: The gray values of PGM2p-GFP protein in the deletion strains DBP7 and YRF1-6 (Lanes 3 and 5) are too high, which will affect the accuracy of quantitative results.
Q3: There was no significant variation in mRNA content between the strains examined (Figure 2B).
What are the average CT values? Please provide corresponding evidence.
Q4: In the conclusion, the author mentioned that this is an important notion as it could have implications in the effectiveness of mRNA vaccines. For their efficacy, mRNA vaccines rely on the host translation machinery to translate the sequences embedded in their mRNAs.
This sentence has nothing to do with the research content and belongs to excessive extension.
Q5: Genetic Interaction Analysis.
This part of the content gives the greatest impression that it is analyzed for the purpose of formal analysis (rather than substantive analysis), rather than being organically integrated with other results.
Reviewer 3 Report
The findings of this research manuscript present concrete mechanism underlying the role of two genes in modulating sensitivity of the cells to lithium Chloride.
The authors used several methods of gene and protein expression, gene interaction, drug sensitivity and quantitative B-galactosidase assay. The results are well-presented, interpreted and conclusion is properly-derived. The text is written in excellent English language.
These findings have a promising potential on the future molecular pharmacokinetics of drugs, used for bipolar disease.
It would be more appropriate, if the authors indicate or suggest, at the end of discussion, that translational or other pre-clinical studies are warranted/needed to see how these mechanistic findings can improve the responders and non-responders bipolar disease patients to Lithium chloride therapy.
